# Estimating the CMIP6 Anthropogenic Aerosol Radiative Effects with the Advantage of Prescribed Aerosol Forcing

**Xiangjun Shi** [1,2,*] , **Chunhan Li** [1], **Lijuan Li** [2], **Wentao Zhang** [1] **and Jiaojiao Liu** [1]

[1] School of Atmospheric Sciences, Nanjing University of Information Science and Technology, Nanjing 210044, China; lichunhan@nuist.edu.cn (C.L.); zwtt@nuist.edu.cn (W.Z.); liujj@nuist.edu.cn (J.L.)

[2] State Key Laboratory of Numerical Modeling for Atmospheric Sciences and Geophysical Fluid Dynamics (LASG), Institute of Atmospheric Physics (IAP), Chinese Academy of Sciences, Beijing 100029, China; ljli@mail.iap.ac.cn

[*] Correspondence: shixj@nuist.edu.cn

**Abstract:** The prescribed anthropogenic aerosol forcing recommended by Coupled Model Intercomparison Project Phase 6 (CMIP6) was implemented in an atmospheric model. With the reduced complexity of anthropogenic aerosol forcing, each component of anthropogenic aerosol effective radiative forcing (ERF) can be estimated by one or more calculation methods, especially for instantaneous radiative forcing (RF) from aerosol–radiation interactions (RFari) and aerosol–cloud interactions (RFaci). Simulation results show that the choice of calculation method might impact the magnitude and reliability of RFari. The RFaci—calculated by double radiation calls—is the definition-based Twomey effect, which previously was impossible to diagnose using the default model with physically based aerosol–cloud interactions. The RFari and RFaci determined from present-day simulations are very robust and can be used as offline simulation results. The robust RFari, RFaci, and corresponding radiative forcing efficiencies (i.e., the impact of environmental properties) are very useful for analyzing anthropogenic aerosol radiative effects. For instance, from 1975 to 2000, both RFari and RFaci showed a clear response to the spatial change of anthropogenic aerosol. The global average RF (RFari + RFaci) has enhanced (more negative) by ~6%, even with a slight decrease in the global average anthropogenic aerosol, and this can be explained by the spatial pattern of radiative forcing efficiency.

**Keywords:** CMIP6 prescribed aerosol forcing; calculation method; radiative forcing efficiency; seasonal variability; spatial change of anthropogenic aerosol

## 1. Introduction

One of the guiding questions of the Coupled Model Intercomparison Project Phase 6 (CMIP6) is "how does the earth system respond to forcing?" [1]. Without reliable forcing estimates, it is very difficult to compare climate model responses to changes in forcing, especially the radiative forcing estimates for short-lived atmospheric aerosols [2,3]. Thus, idealized experiments have been designed to highlight and understand the differences in climate model responses to specified common anthropogenic aerosol forcing [4,5]. Given these concepts, a simple plume implementation of the second version of the Max Planck Institute Aerosol Climatology (MACv2-SP) was developed for climate models, which provides prescribed anthropogenic aerosol optical properties and normalized changes in cloud droplet number [6,7].

The anthropogenic aerosol effects on the planetary energy balance can be expressed as effective radiative forcing (ERF), which can be decomposed into the contributions of instantaneous radiative forcing (RF) and rapid adjustments (i.e., ERF − RF) [8–11]. As the complexity of anthropogenic aerosol forcing decreases (i.e., MACv2-SP), the RF from anthropogenic aerosol–cloud interactions (RFaci) and the RF from anthropogenic aerosol–radiation interactions (RFari) can be calculated by double radiation calls (i.e.,

with and without anthropogenic aerosol forcing) at each model time step. In very early climate models with prescribed aerosol and simple cloud microphysics schemes, the RFaci—also known as the Twomey effect—used to be a standard diagnostic (e.g., [12]). However, with the development of climate models, it became very difficult to diagnose RFaci when aerosol–cloud interactions were treated by physically based schemes because the instantaneous RFaci was mixed with its subsequent rapid adjustments. As a result, the RFaci was considered as a theoretical construct and was not quantified in the Cloud and Aerosol chapter of the Intergovernmental Panel on Climate Change (IPCC) Fifth Assessment Report (AR5) [8]. Fortunately, MACv2-SP makes it possible to directly diagnose RFaci once again. Another benefit of using MACv2-SP is that only one present-day (PD) simulation can provide an estimate of RFari by double radiation calls with and without anthropogenic aerosol optical properties [7,13]. It should be noted that when using climate models that treat anthropogenic aerosol processes in detail, both PD and preindustrial (PI) simulations are needed for calculating anthropogenic aerosol RFari because it is difficult to isolate anthropogenic aerosols from natural aerosols [9]. This calculation method (i.e., the difference between two simulations) also works for climate models with MACv2-SP. In short, with the advantage of prescribed forcing, each basic component of anthropogenic aerosol ERF (i.e., RFari, RFaci, and rapid adjustments) can be estimated by one or more calculation methods.

The perturbation of anthropogenic aerosol ERF is considerable, owing to the model internal year-to-year variability, while the RF is stable. Fortunately, the RF contributes most to the ERF [6,7,13,14]. This study shows that both RFari and RFaci determined from a PD simulation with prescribed anthropogenic aerosol forcing are very robust and can be used as offline simulation results. The seasonal variabilities of the robust RF (i.e., RFari and RFaci) can be analyzed based on climate model simulations. It should be noted that, unlike the robust RF, the ERF in one season is the response to the aerosol forcing from all seasons. Thus, no studies have been conducted that show the seasonal variability of ERF. Another benefit of the robust RF is that the changes in RFari and RFaci caused by the spatial shift in anthropogenic aerosol between the 1970s and 2000s should be very clear. The spatial shift in anthropogenic aerosol emissions (e.g., sulfur dioxide) between the 1970s and 2000s is notable due to the increased emissions in China [15]. However, a previous study suggested that the changes in ERF caused by this spatial shift were diverse both in terms of spatial structure and global average [16]. Even if using MACv2-SP, model simulations have shown little change in global average anthropogenic aerosol ERF and this small change was difficult to detect, owing to the notable model internal year-to-year variability [6,13]. In short, with the advantage of prescribed aerosol forcing, the anthropogenic aerosol radiative effects can be estimated in terms of the robust RFari and RFaci.

In this study, the MACv2-SP was implemented into the Grid-point Atmospheric Model of IAP LASG (GAMIL), which was used in the CMIP6 as the atmospheric component of the Flexible Global Ocean–Atmosphere–Land System Model developed by the Chinese Academy of Sciences (CAS-FGOALS) [17]. Meanwhile, the diagnostic package of this climate model was improved to calculate the RFari and RFaci by double radiation calls. In order to demonstrate the advantages of prescribed anthropogenic aerosol forcing (i.e., MACv2-SP), our estimates focus on the following aspects: Firstly, the basic components of ERF are estimated, in particular, for RFari and RFaci. As part of this, the diversity in estimates stemming from different calculation methods are discussed. Secondly, the seasonal variabilities of anthropogenic aerosol RF (i.e., RFari and RFaci) are presented. The corresponding radiative forcing efficiencies, which indicate the impact of environmental properties on the RF, are also analyzed. Finally, the impact of the spatial shift in anthropogenic aerosol between the 1970s and 2000s is investigated. The paper is organized as follows: The calculation methods and experimental design are described in Section 2; the simulation results are presented and analyzed in Section 3; and finally, conclusions and discussion are provided in Section 4.

## 2. Methods

### 2.1. The GAMIL Model with MACv2-SP

Stevens et al. (2017) provided details about MACv2-SP [5]. Nine plumes (five industrial and four biomass) are used to capture the spatial distribution of the anthropogenic aerosol in the MACv2-SP. The spatial structure (horizontal structure and vertical structure) and annual cycle of each plume are parameterized as a basis function to represent the anthropogenic aerosol. Here, a brief introduction is provided to better understand the radiative forcing variables used in this study. To represent anthropogenic aerosol–radiation interactions, MACv2-SP provides anthropogenic aerosol optical properties (i.e., the aerosol optical depth (AOD), single-scattering albedo, and asymmetry factor). In the radiation scheme, total aerosol optical properties are calculated based on the anthropogenic aerosol optical properties from MACv2-SP and natural aerosol optical properties from the default mechanism of the host model. As only smaller, fine-mode aerosol contributes to MACv2-SP, only shortwave anthropogenic aerosol optical properties are provided. To represent the anthropogenic aerosol Twomey effect, MACv2-SP provides a normalized change in cloud droplet number (rNc). The rNc is the increasing ratio of cloud droplet number (Nc) as compared to the host model background Nc (i.e., only natural aerosols contribute). In the year 1850 (pre-industrial times, PI), there was no anthropogenic aerosol forcing in MACv2-SP, and so the rNc is taken as a constant, 1. After 1850, the rNc (>1) is used to tune the host model Nc and ensure that the proportional change in Nc caused by anthropogenic aerosol is insensitive to background Nc. Notably, only the warm cloud Twomey effect, which refers to cloud optical thickness under a fixed liquid water content increased by anthropogenic aerosol [18], is considered in the design of MACv2-SP. Based on this conception, the rNc is only used for calculating warm cloud optical properties in the radiation scheme. In the cloud microphysics scheme, the Nc is not affected by the rNc (i.e., no cloud-lifetime effect).

The GAMIL model is a Grid-point Atmospheric general circulation Model of IAP LASG with a finite difference dynamical core, developed by the Institute of Atmospheric Physics, Chinese Academy of Sciences [17,19–23]. Details of the GAMIL model can be found in the study of Li et al. 2020 [17]. Radiative transfer is solved by delta-Eddington approximation with 19 solar spectral bands [24,25]. In the default GAMIL model, the natural aerosol direct radiative effect is represented by given natural aerosol optical properties. Here, anthropogenic aerosol optical properties from MACv2-SP were superimposed. In the default GAMIL model, the indirect aerosol effects are treated by a detailed two-moment cloud microphysics scheme [23]. In all experiments in this study, the Nc used in the cloud microphysics scheme was calculated based on a dataset of prescribed PI aerosols (i.e., natural aerosol), which includes sulfate, hydrophobic black carbon, hydrophilic black carbon, hydrophobic organic carbon, hydrophilic organic carbon, dust, and sea salt. For considering the anthropogenic aerosol Twomey effect provided by MACv2-SP (i.e., rNc), the warm cloud optical properties used in the radiation scheme were calculated by Nc × rNc, instead of Nc.

### 2.2. Calculation Method and Updated Diagnostic Package

There is no perfect method to determine the ERF. One recommended way to calculate the ERF is by using the radiative flux perturbation method from the top-of-the-atmosphere (TOA) energy balance difference between two simulations with and without anthropogenic aerosol but the same sea surface temperature (SST) [8–10]. In this study, this fixed-SST method was used to calculate the ERF. The RF can be calculated by double radiation calls in the model diagnostic package, and the rapid adjustments can be diagnosed as ERF − RF [7,13].

Using GAMIL with MACv2-SP, the anthropogenic aerosol ERF from combined aerosol–radiation and aerosol–cloud interactions (also named as ERFari + aci) can be decomposed into the anthropogenic aerosol RFari (i.e., instantaneous direct effect), the anthropogenic aerosol RFaci (i.e., instantaneous Twomey effect), and rapid adjustments (i.e., ERFari + aci − RFari − RFaci). Notably, these rapid adjustments cannot be decomposed

into the rapid adjustments from aerosol–radiation interactions and the rapid adjustments from aerosol–cloud interactions because all adjustments interact with each other at each model time step. The ERF (short for anthropogenic aerosol ERF) from two simulations with and without anthropogenic aerosol–radiation interactions only (ERFari) can be decomposed into the RFari (short for anthropogenic aerosol RFari) and corresponding rapid adjustments from aerosol–radiation interactions. Among these rapid adjustments, the rapid adjustment induced by the changes in clouds is referred to as the semi-direct effect [8,9]. The ERF from aerosol–cloud interactions only (ERFaci) can be decomposed into the RFaci (short for anthropogenic aerosol RFaci) and corresponding rapid adjustments (no lifetime effect).

In order to use several possible calculation methods to diagnose RFari and RFaci, the radiation subroutine needs to be called many times to provide different net radiative fluxes at the TOA. All shortwave net radiative fluxes diagnosed from the radiation scheme are listed in Table 1. For the convenience of remembering, these variables are named according to a certain rule. The capital letter "F" denotes the benchmark of shortwave net radiative fluxes at the TOA, which excludes the radiative effects of clouds and aerosols (both anthropogenic aerosol and natural aerosol). The subscript letters ("A", "a", "C", or "Cc") indicate the radiative forcing factor considered in the radiation transfer calculation, as compared to the benchmark "F". The letters correspond to: natural aerosol optical properties ("A"); anthropogenic aerosol optical properties ("a"); background cloud optical properties ("C"); cloud optical properties with the Twomey effect ("Cc"). The "C" was calculated based on Nc, and the "Cc" based on Nc × rNc. Here, we introduce a few variables that are listed in Table 1. According to the naming rule, $F_{AaCc}$ considers all radiative forcing factors, which is the commonly called the shortwave net radiative flux. $F_{Aa}$ is clear-sky $F_{AaCc}$. Compared to $F_{AaCc}$, $F_{AaC}$ excludes the anthropogenic aerosol Twomey effect ("c"). Compared to $F_{AaCc}$, $F_{ACc}$ excludes the anthropogenic aerosol direct radiative effect ("a"). Compared to $F_{AaCc}$, $F_{Cc}$ excludes the total (natural and anthropogenic) aerosol direct radiative effect ("A" and "a").

**Table 1.** Different shortwave net radiative fluxes at the top-of-the-atmosphere (TOA).

| Names (W m$^{-2}$) | Natural Aerosol Optical Properties ("A") | Anthropogenic Aerosol Optical Properties ("a") | Cloud Optical Properties with the Twomey Effect ("Cc") | Background Cloud Optical Properties ("C") |
|---|---|---|---|---|
| $F_{AaCc}$ | X | X | X | |
| $F_{Aa}$ | X | X | | |
| $F_{ACc}$ | X | | X | |
| $F_A$ | X | | | |
| $F_{aCc}$ | | X | X | |
| $F_a$ | | X | | |
| $F_{Cc}$ | | | X | |
| $F$ | | | | |
| $F_{AaC}$ | X | X | | X |
| $F_{AC}$ | X | | | X |

Model diagnostic instantaneous radiative forcing variables were calculated as the difference between two net radiative fluxes listed in Table 1. Table 2 lists the variables analyzed in this study. These variables are also named according to a certain rule. Anthropogenic aerosol ("a") direct radiative forcing is named as $aF_{XX}$. Total (natural and anthropogenic) aerosol ("A" and "a") direct radiative forcing is named as $AaF_{XX}$. Cloud ("Cc", with the Twomey effect) forcing is named as $CcF_{XX}$. Background cloud ("C", without the Twomey effect) forcing is named as $CF_{XX}$. The instantaneous Twomey effect ("c") is named $cF_{xx}$. Here, the subscript "XX" refers to background radiative forcing factors (i.e., "A", "a", "C", and "Cc") that are considered in the radiation transfer. For instance, both $aF_{ACc}$ ($aF_{Acc} = F_{AaCc} - F_{ACc}$) and $aF_{Cc}$ ($aF_{Cc} = F_{aCc} - F_{Cc}$) indicate anthropogenic aerosol

("a") radiative forcing. Compared to $aF_{ACc}$, $aF_{Cc}$ excludes the impact of the natural aerosol direct radiative effect ("A"). Notably, all $aF_{XX}$ indicate RFari, and all $cF_{XX}$ indicate RFaci.

**Table 2.** List of model diagnostic instantaneous radiative forcing (RF) variables.

| Names | Description | Equation |
|---|---|---|
| $AaF_{Cc}$ | Shortwave total aerosol RF | $AaF_{Cc} = F_{AaCc} - F_C$ |
| $AaF$ | Clear-sky $AaF_{Cc}$ | $AaF = F_{Aa} - F$ |
| $aF_{ACc}$ | Shortwave anthropogenic aerosol RF | $aF_{ACc} = F_{AaCc} - F_{ACc}$. |
| $aF_A$ | Clear-sky $aF_{ACc}$ | $aF_A = F_{Aa} - F_A$ |
| $aF_{Cc}$ | $aF_{ACc}$ without natural aerosol radiative effect | $aF_{Cc} = F_{aCc} - F_{Cc}$. |
| $aF$ | Clear-sky $aF_{Cc}$ | $aF = F_a - F$ |
| $aF_{AC}$ | $aF_{ACc}$ excluding Twomey effect | $aF_{AC} = F_{AaC} - F_{AC}$. |
| $aF_{Cc}dA$ | Impact of natural aerosol on calculating $aF_{Cc}$ | $aF_{Cc}dA = aF_{ACc} - aF_{Cc}$ |
| $aFdA$ | Clear-sky $aF_{Cc}dA$ | $aFdA = aF_A - aF$ |
| $CcF_{Aa}$ | Shortwave cloud forcing | $CcF_{Aa} = F_{AaCc} - F_{Aa}$ |
| $CcF$ | $CcF_{Aa}$ without total aerosol radiative effect | $CcF = F_{Cc} - F$ |
| $CcF_A$ | $CcF_{Aa}$ without anthropogenic aerosol radiative effect | $CcF_A = F_{ACc} - F_A$ |
| $CF_{Aa}$ | $CcF_{Aa}$ excluding Twomey effect | $CF_{Aa} = F_{AaC} - F_{Aa}$ |
| $CF_A$ | $CcF_A$ excluding Twomey effect | $CF_A = F_{AC} - F_A$ |
| $cF_{AaC}$ | Twomey effect on cloud forcing | $cF_{AaC} = CcF_{Aa} - CF_{Aa} = F_{AaCc} - F_{AaC}$ |
| $cF_{AC}$ | $cF_{AaC}$ without anthropogenic aerosol radiative effect | $cF_{AC} = CcF_A - CF_A = F_{ACc} - F_{AC}$ |

*2.3. Experimental Design*

In order to show each component of the ERF and the impact of the spatial shift in anthropogenic aerosol, five experiments—referred to as BASE, RAD, TMY, ALL, and PAT—were carried out in this study. The experimental setups are summarized in Table 3. The natural aerosol direct and indirect effects were considered in all the experiments. The BASE experiment did not consider anthropogenic aerosol forcing. In contrast to the BASE experiment, the PD (year 2000) anthropogenic aerosol radiative effect and Twomey effect provided by MACv2-SP were added in the ALL experiment, the RAD experiment only added the anthropogenic aerosol (year 2000) direct radiative effect, and the TMY experiment only added the Twomey effect (year 2000). The PAT experiment (PAT is short for spatial pattern) was similar to the ALL experiment, but the year 1975 was used for calculating the anthropogenic aerosol forcing. All experiments were atmosphere-only simulations with same prescribed climatological ocean surface conditions. Anthropogenic aerosol data calculated from MACv2-SP during a given year were used in the model simulations, which did not change from year to year. All simulations were run for 11 model years at a horizontal grid resolution of 80 × 180 and 26 vertical levels. It is noteworthy that the TOA (i.e., the second top interface level, ~2.6 hPa) is different from the top of the model (i.e., the first top interface level, 0). The first year was considered as a spin-up period and not included in the analysis. The standard deviations, which were estimated from the averages of each year (i.e., 10 averages), were used for variability analysis.

**Table 3.** List of experiments.

| Names | Anthropogenic Aerosol Radiative Forcing ("a") | Anthropogenic Aerosol Twomey Effect ("c") |
|---|---|---|
| BASE | None | None |
| RAD | Year 2000 | None |
| TMY | None | Year 2000 |
| ALL | Year 2000 | Year 2000 |
| PAT | Year 1975 | Year 1975 |

## 3. Results

Because only shortwave anthropogenic aerosol optical properties are provided by MACv2-SP, only shortwave radiative forcing variables are analyzed. When analyzing a radiative variable, it is necessary to know which experiment it comes from. To show the source of the variable, the experiment name is added in superscript. For example, the $AaF_{Cc}$ from the ALL experiment is denoted as $AaF_{Cc}^{ALL}$, and the difference in $AaF_{Cc}$ between the ALL and BASE experiments is denoted as $AaF_{Cc}^{ALL-BASE}$. Furthermore, for ease of expression, "△" is used to denote the difference between two simulations (e.g., $\Delta AaF_{Cc}$). For convenience of searching and comparison, the global annual mean radiative variables from all experiments are listed in Table A1. Based on Table A1, the different calculation methods for estimating anthropogenic aerosol radiative effects are summarized in the Appendix A.

### 3.1. Annual Mean Results

Figure 1 shows the all-sky and clear-sky RFari derived from two kinds of calculation methods. The RFari from the difference between two simulations ($AaF_{Cc}^{RAD-BASE}$) and the RFari from one simulation ($aF_{ACc}^{RAD}$) show the same global cooling at $-0.21$ W m$^{-2}$. It should be noted that all regions of RFari ($aF_{ACc}^{RAD}$) are statistically significant, whereas RFari ($AaF_{Cc}^{RAD-BASE}$) is not statistically significant over some regions. RFari is usually more negative under clear-sky conditions than under all-sky conditions [26,27]. As expected, both $AaF^{RAD-BASE}$ and $aF_A^{RAD}$ (i.e., clear-sky RFari) show a stronger (more negative) global cooling at $-0.45$ W m$^{-2}$. Compared to all-sky RFari ($AaF_{Cc}^{RAD-BASE}$), clear-sky RFari ($AaF^{RAD-BASE}$) has more statistically significant regions. This result can be attributed to the fact that the difference in cloud optical properties between the RAD and BASE experiments is excluded. Unlike $aF_A^{RAD}$, there are still some statistically non-significant regions of $AaF^{RAD-BASE}$ due to model internal year-to-year variability. The comparison between $AaF_{Cc}^{RAD-BASE}$ ($AaF^{RAD-BASE}$) and $aF_{ACc}^{RAD}$ ($aF_A^{RAD}$) indicates that the RFari estimated by a model diagnostic radiative forcing variable from one simulation is more robust than that estimated by the difference between two simulations. The $aF_{ACc}^{RAD}$ and $aF_A^{RAD}$ are scarcely affected by model internal year-to-year variability and can be considered as offline simulation results. There are many other methods for calculating RFari (e.g., $AaF_{Cc}^{ALL-TMY}$, $AaF_{Cc}^{ALL-BASE}$, and $aF_{ACc}^{ALL}$). It is necessary to point out that $AaF_{Cc}^{ALL-TMY}$ and $AaF_{Cc}^{ALL-BASE}$ are similar to $AaF_{Cc}^{RAD-BASE}$, and $aF_{ACc}^{ALL}$ is almost the same as $aF_{ACc}^{RAD}$ (not shown). Both all-sky RFari and clear-sky RFari show some regions with a warming effect (Figure 1). Kinne (2019) explained this warming effect by dimming over snow and lower clouds [26]. Taking the snow-covered regions as an example, a relatively larger proportion of downward solar radiation is reflected by the surface (not shown), and thus, the role of anthropogenic aerosols in absorbing and scattering the reflected solar radiation (warming effect) becomes relatively important.

We found that the GAMIL model shows a very strong all-sky and clear-sky natural aerosol RF ($-5.74$ and $-8.58$ W m$^{-2}$, $AaF_{Cc}^{BASE}$ and $AaF^{BASE}$ in Table A1), which has an obvious impact on calculating the all-sky and clear-sky RFari (0.12 and 0.19 W m$^{-2}$, $aF_{Cc}dA$ and $aFdA$ in Table A1). Therefore, it is necessary to analyze the RFaci without the impact of natural aerosol. Taking the results from the RAD experiment as an example to explain (Figure 2), $aF_{Cc}^{RAD}$ excludes the natural aerosol impact as compared to $aF_{ACc}^{RAD}$. The $aF_{Cc}dA^{RAD}$ ($aF_{Cc}dA^{RAD} = aF_{ACc}^{RAD} - aF_{Cc}^{RAD}$) indicates the impact of natural aerosols on estimating RFari. The $aF^{RAD}$ and $aFdA^{RAD}$ are the clear-sky $aF_{Cc}^{RAD}$ and $aF_{Cc}dA^{RAD}$, respectively. Without the impact of natural aerosols, the global average values of $aF_{Cc}^{RAD}$ and $aF^{RAD}$ are $-0.33$ and $-0.64$ W m$^{-2}$, respectively. The global mean $aF_{Cc}dA^{RAD}$ and $aFdA^{RAD}$ is 0.12 and 0.19 W m$^{-2}$, respectively, and these values are approximately one-third of the RFari strength without the influence of natural aerosols (i.e., $aF_{Cc}^{RAD}$ and $aF^{RAD}$). These results suggest that the natural aerosol radiative effect might have an obvious impact on estimating RFari. The model diversity in estimating RFari might stem from the difference in the host model natural aerosol.

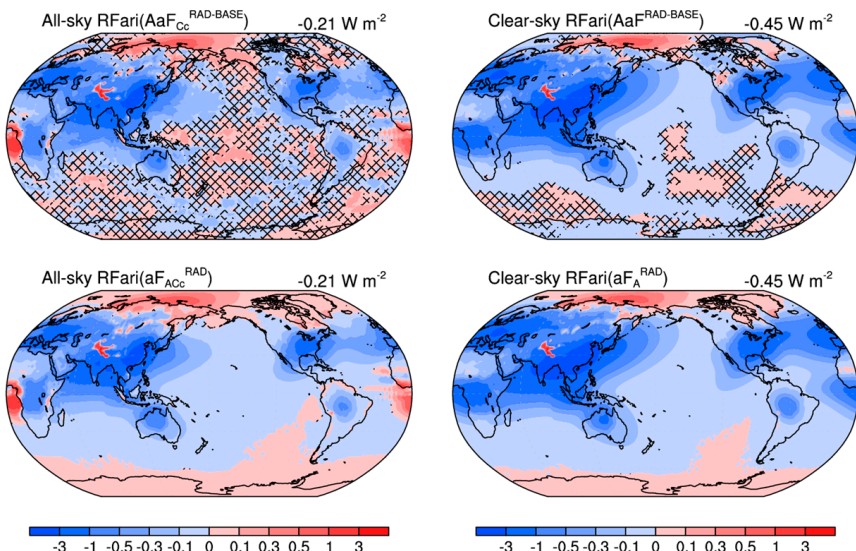

**Figure 1.** Annual mean maps for the present-day (year 2000) anthropogenic aerosol all-sky (**left**) and clear-sky (**right**) instantaneous radiative forcing from aerosol–radiation interactions (RFari). The results calculated as the differences between the RAD and BASE experiments are shown in the upper panels. The results provided by the RAD experiment only are shown in the lower panels. The global average is given in the upper-right corner. Hatching represents the nonsignificant area at the 90% confidence level of Student's *t*-test.

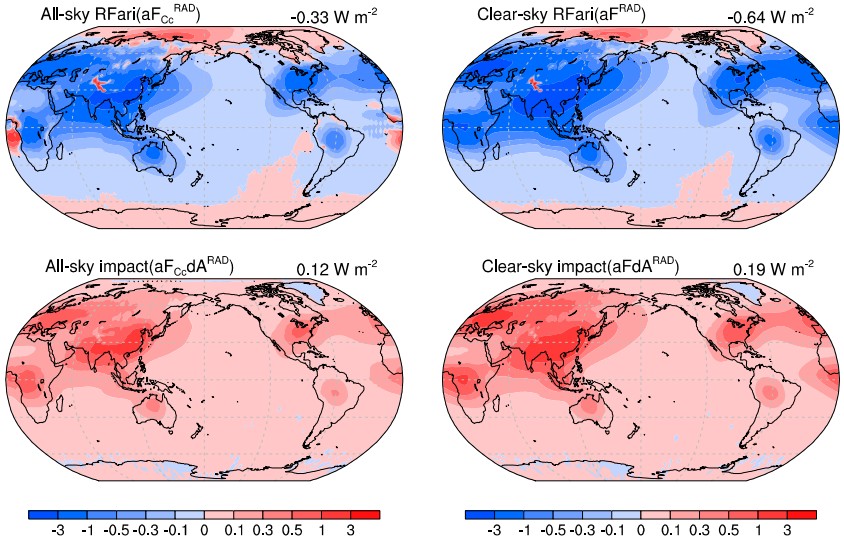

**Figure 2.** As in Figure 1 but for the all-sky (**left**) and clear-sky (**right**) instantaneous direct radiative effect without the natural aerosol radiative effect (RFari, upper panels) and the impact of the natural aerosol radiative effect on calculating RFari (lower panels).

This paragraph analyzes the anthropogenic aerosol semi-direct effect (Figure 3). Both $CF_A^{RAD-BASE}$ and $CF_A^{ALL-TMY}$ represent the semi-direct effect. $CF_A^{RAD-BASE}$ shows global cooling at $-0.01$ W m$^{-2}$, whereas $CF_A^{ALL-TMY}$ shows global warming at 0.12 W m$^{-2}$. Their difference (0.13 W m$^{-2}$) is less than their corresponding standard deviations, which are estimated from the global averages of each year (0.18 and 0.15 W m$^{-2}$, Table A1). It is noteworthy that the standard deviations listed in Table A1 indicate the year-to-year variability of the global average, and those shown in Figure 3 represent the year-to-year variability of every model grid. Thus, the global averages of the standard deviations shown

in Figure 3 (4.06 and 4.12 W m$^{-2}$) are dozens of times larger than those listed in Table A1. For a local region, the semi-direct effect standard deviation is usually much larger than the 10-year average. This suggests that it is very difficult for a local region to yield a reliable multi-year average of the semi-direct effect, even with a long-term simulation (e.g.,100-year simulation). The standard deviation from $CF_A^{RAD-BASE}$ is similar to that from $CF_A^{ALL-TMY}$, and this result indicates that the magnitude of the year-to-year variability (i.e., the standard deviation) of the semi-direct effect is relatively stable. There are many other methods for calculating the semi-direct effect (e.g., $CcF_A^{RAD-BASE}$, $CcF^{RAD-BASE}$, $CcF_A^{ALL-TMY}$ and $CcF^{ALL-TMY}$). It is necessary to point out that $CcF_A^{RAD-BASE}$ and $CcF^{RAD-BASE}$ are similar to $CF_A^{RAD-BASE}$, and $CcF_A^{ALL-TMY}$ and $CcF^{ALL-TMY}$ are similar to $CF_A^{ALL-TMY}$ (not shown).

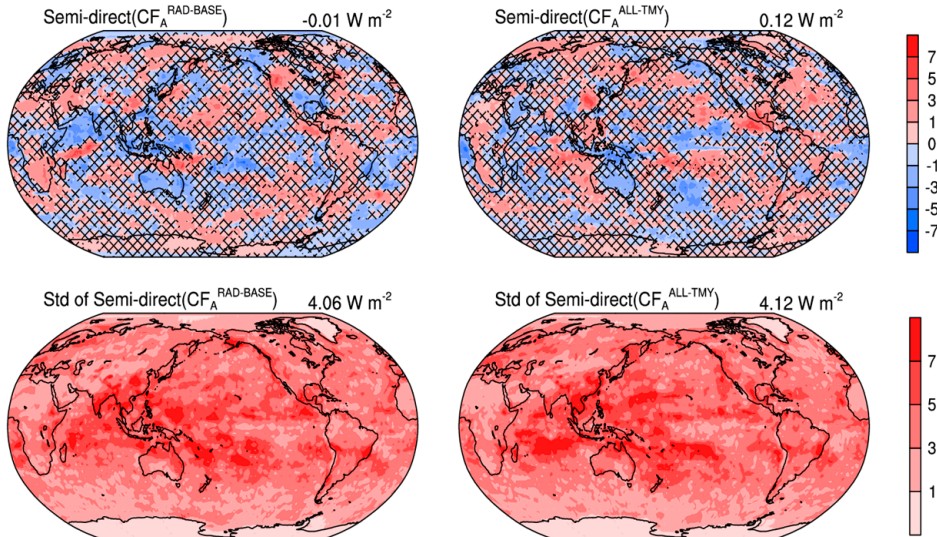

**Figure 3.** Annual mean anthropogenic aerosol (year 2000) semi-direct effect (**upper panels**) and corresponding standard deviations calculated from the difference of each year (**lower panels**). The results calculated from the differences between the RAD and BASE experiments are presented in the left-hand column. The results calculated from the differences between the ALL and TMY experiments are presented in the right-hand column. The global average is given in the upper-right corner. Hatching represents the nonsignificant area at the 90% confidence level of Student's *t*-test. Note: that the colorbar of the radiation variable is different from that in the other figures.

This paragraph analyzes anthropogenic aerosol indirect effects. Both $\Delta CcF$ and $\Delta CcF_A$ can be used to quantify aerosol indirect effects on warm clouds induced by the Twomey effect [9]. Here, in order to exclude semi-direct effect, "$\Delta$" only denotes the difference between two simulations with and without Twomey effect (i.e., TMY − BASE and ALL − RAD). $CcF^{TMY-BASE}$ is almost the same as $CcF_A^{TMY-BASE}$, and $CcF^{ALL-RAD}$ is almost the same as $CcF_A^{ALL-RAD}$ (not shown). This suggests that the impact of natural aerosol on estimating aerosol indirect effects is negligible. Here, only $\Delta CcF_A$ (i.e., $CcF_A^{TMY-BASE}$ and $CcF_A^{ALL-RAD}$) are analyzed (Figure 4). The $\Delta CcF_A$ can be decomposed into $\Delta cF_{AC}$ and $\Delta CF_A$ [$\Delta CcF_A = \Delta(CcF_A - CF_A) + \Delta CF_A = \Delta cF_{AC} + \Delta CF_A$]. Because both $cF_{AC}^{BASE}$ and $cF_{AC}^{RAD}$ are zero, and both $cF_{AC}^{TMY}$ and $cF_{AC}^{ALL}$ denote the instantaneous Twomey effect (i.e., RFaci). In other words, the anthropogenic aerosol indirect effects (i.e., $CcF_A^{TMY-BASE}$ and $CcF_A^{ALL-RAD}$) can be decomposed into the instantaneous Twomey effect (i.e., $cF_{AC}^{TMY}$ and $cF_{AC}^{ALL}$) and subsequent changes in cloud forcing induced by the Twomey effect ($CF_A^{TMY-BASE}$ and $CF_A^{ALL-RAD}$). Figure 4 shows these variables. Both $cF_{AC}^{TMY}$ and $cF_{AC}^{ALL}$ give a global average RFaci of −0.10 W m$^{-2}$. All regions of $cF_{AC}^{TMY}$ and $cF_{AC}^{ALL}$ are statistically significant, and $cF_{AC}^{TMY}$ is almost the same as $cF_{AC}^{ALL}$. This result indicates that the simulated RFaci (i.e., instantaneous Twomey effect) is very robust. The rapid adjustment in cloud forcing estimated by $CF_A^{TMY-BASE}$ shows global cooling at −0.07 W m$^{-2}$, whereas $CF_A^{ALL-RAD}$ shows global warming at 0.06 W m$^{-2}$. It should

be noted that their difference (0.13 W m$^{-2}$) is less than their standard deviations of the global average (0.16 and 0.17 W m$^{-2}$, Table A1). Furthermore, although both $CF_A^{TMY-BASE}$ and $CF_A^{ALL-RAD}$ show that there are a few statistically significant regions, the comparison between $CF_A^{TMY-BASE}$ and $CF_A^{ALL-RAD}$ shows that these statistically significant regions are not fixed. In short, the rapid adjustment in cloud forcing (i.e., $\Delta CF_A$) is obviously affected by model internal year-to-year variability. It is clear that the perturbation of the anthropogenic aerosol indirect effects ($\Delta CcF_A$, upper panels of Figure 4) depends almost entirely on its rapid adjustment ($\Delta CF_A$, lower panels of Figure 4). Similarly to the semi-direct effect, it is very difficult for a local region to yield a reliable multi-year average of $\Delta CF_A$. Therefore, it is necessary to decompose anthropogenic aerosol indirect effects into a robust instantaneous Twomey effect (i.e., definition-based Twomey effect) and unstable subsequent changes in cloud forcing induced by the Twomey effect. With the benefit of this decomposition, the model intercomparison study can focus on the robust instantaneous Twomey effect.

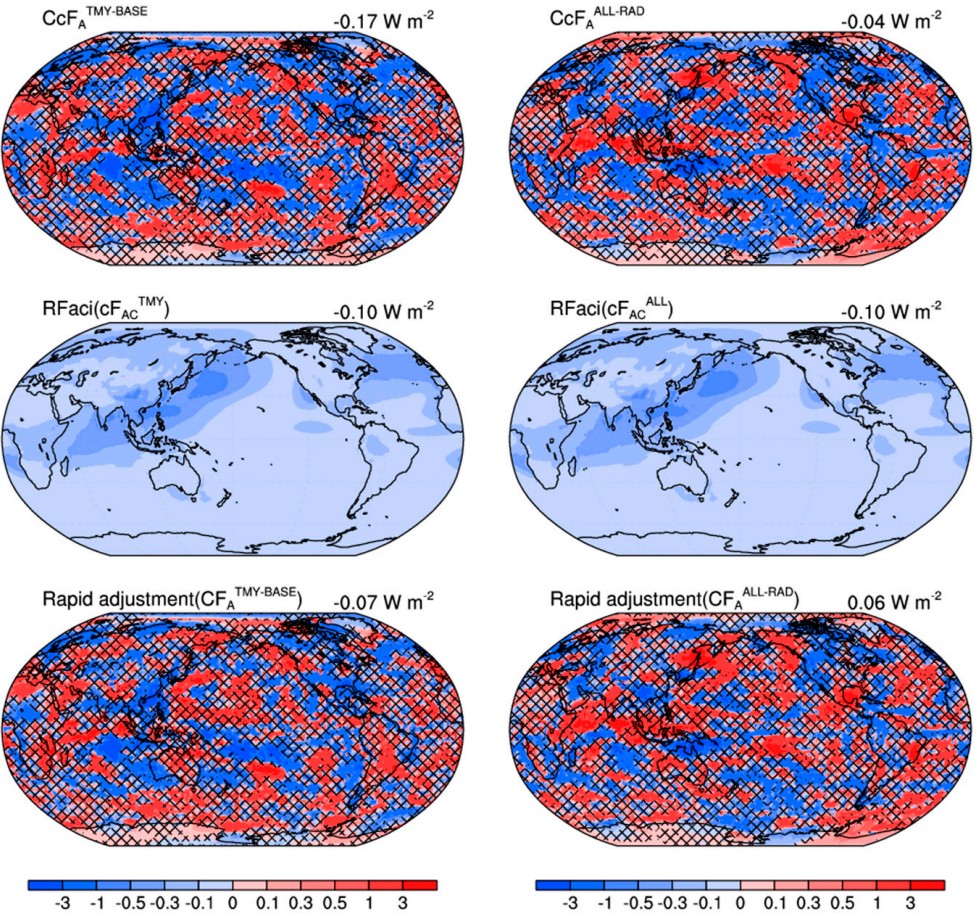

**Figure 4.** Annual mean anthropogenic aerosol (year 2000) Twomey effect (change in cloud forcing, **upper panels**) and its two components: the instantaneous Twomey effect (RFaci, **middle panels**) and corresponding rapid adjustment (**lower panels**). The results determined from the TMY and BASE experiments are presented in the left-hand column. The results determined from the RAD and ALL experiments are presented in the right-hand column. The global average is given in the upper-right corner. Hatching represents the nonsignificant area at the 90% confidence level of Student's *t*-test.

### 3.2. Seasonal Variability

Figure 5 compares the seasonal variations in anthropogenic AOD, RFari, and corresponding radiative forcing efficiency (RFari/AOD). The radiative forcing efficiency (hereafter "efficiency") is used to indicate the impact of environmental properties (such

as surface albedo, solar insolation and even clouds) on the RFari [26]. Anthropogenic AOD is highest in the Northern Hemisphere summer (0.037) and lowest in winter (0.027). Both $aF_{ACc}^{RAD}$ (with the impact of natural aerosol) and $aF_{Cc}^{RAD}$ (without the impact of natural aerosol) show the strongest RFari in summer (−0.25 and −0.40 W m$^{-2}$) and the weakest (less negative) RFari in winter (−0.15 and −0.26 W m$^{-2}$). The efficiency also shows notable seasonal variations. In the Northern Hemisphere, there is more snow cover and less sunshine during the winter season. As expected, the global average efficiency is weakest in winter. This is also a reason for the weakest RFari during the winter season. In the Southeast Asia region, the efficiency in summer is weaker than that in other seasons. This might be caused by there being relatively more clouds during the summer season. Under this influence, the global mean efficiency is strongest in autumn, not in summer. The global average efficiency without the influence of natural aerosol is approximately −10 W m$^{-2}$ per unit anthropogenic AOD, and this value is close to the estimate (−12 W m$^{-2}$ per unit AOD) from an offline radiative transfer model with MACv2 [26].

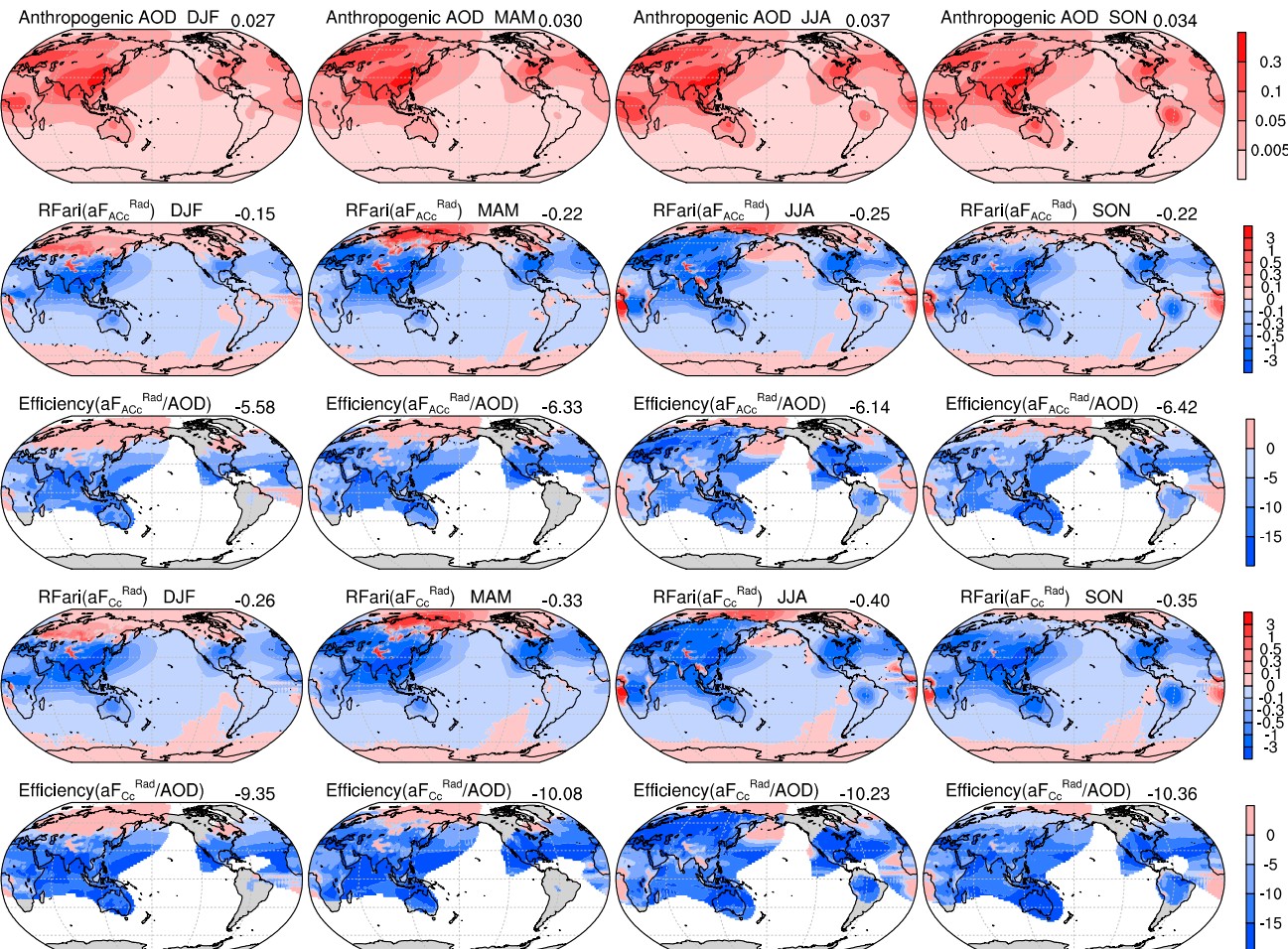

**Figure 5.** Seasonal maps for the present-day (year 2000) anthropogenic aerosol optical depth (AOD) (**top** row), the normal instantaneous radiative forcing from aerosol–radiation interactions (RFari = $aF_{ACc}$, **second** row) and corresponding efficiency (**third** row), and the RFari without the natural aerosol radiative effect (RFari = $aF_{Cc}$, **fourth** row) and corresponding efficiency (bottom row). Season labels (DJF, MAM, JJA, and SON) are given at the top. The area of low anthropogenic aerosol burden (AOD < 0.005) is masked.

The seasonal variability of the anthropogenic aerosol Twomey effect is presented in Figure 6. Consistent with anthropogenic AOD (Figure 5, top row), the droplet number increasing factor (rNc) is largest in the Northern Hemisphere in summer (1.077) and smallest in winter (1.071). Regionally, rNc peaks over East Asia in every season. The global

average RFaci ($cF_{AC}^{TMY}$) in winter, spring, summer, and autumn are −0.06, −0.10, −0.15, and −0.09 W m$^{-2}$, respectively. The seasonal variability of RFaci is stronger than that of RFari. Similar to the efficiency for RFari, the efficiency for the Twomey effect was calculated by RFaci / (rNc −1). The RFaci efficiency also shows notable seasonal variations. In every season, the RFaci efficiency over the ocean is usually larger than that over the land—a result that is in agreement with the spatial distribution of shortwave cloud forcing (not shown). In other words, the RFaci efficiency usually enhances (more negative) with increasing cloud forcing. In the East Asia region (i.e., an area with a high anthropogenic aerosol burden), shortwave cloud forcing is strongest in summer (not shown). As a result, the RFaci efficiency is also strongest in summer. It should be noted that, in the East Asia region, the RFari efficiency is weakest in summer, owing to stronger cloud forcing (Figure 5). This is the reason why the seasonal variability of RFaci is stronger than that of RFari.

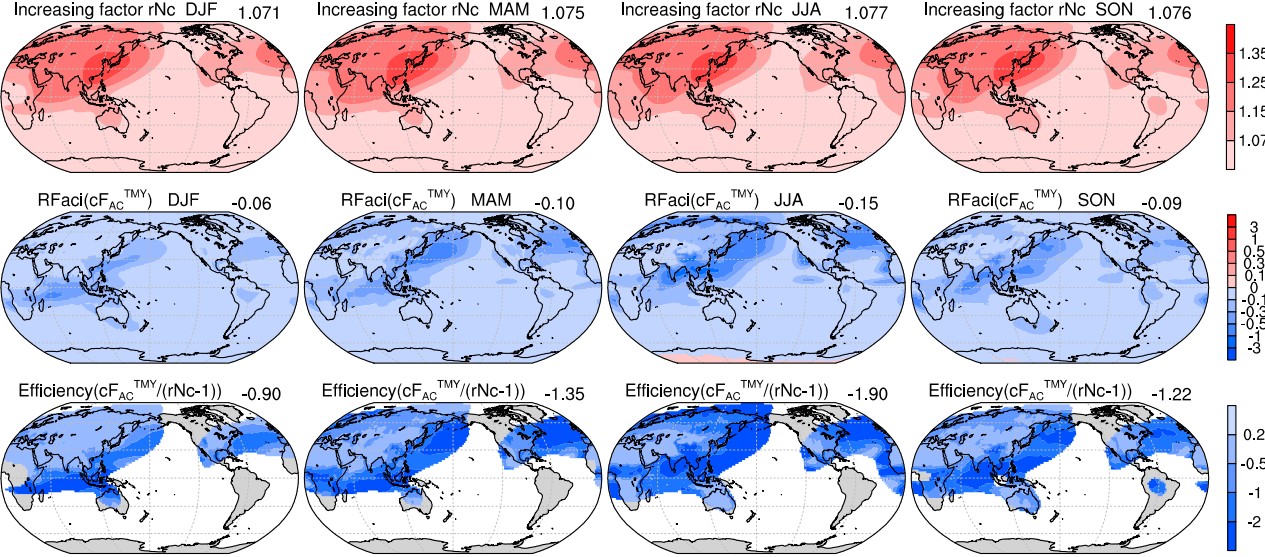

**Figure 6.** As in Figure 5 but for the present-day (year 2000) droplet number increasing factor (rNc, **upper panels**), the instantaneous radiative forcing from aerosol–cloud interactions (RFaci, **middle panels**), and the efficiency for aerosol–cloud interactions (**lower panels**). The area with a low anthropogenic aerosol burden (rNc < 1.07) is masked.

### 3.3. Impact of Spatial Distributions

Figure 7 shows the anthropogenic aerosol forcing data used in this study. The global average anthropogenic AOD in 2000 and 1975 are almost the same (0.032). Their difference (2000−1975) is −0.0007. This difference indicates that the global average AOD in the year 2000 is slightly less than that in 1975. Notably, there have been a few studies that estimated anthropogenic aerosol forcing based on MACv2-SP (e.g., [13,26]), and the year 2005 was chosen as the PD in these studies. However, in our study, the year 2000 was chosen as the PD rather than 2005, because the global average AOD in 2005 was 0.034 (not shown), which is obviously greater than that in 1975. In this study, the conclusion about the spatial pattern can be better explained if the global average AOD from the PD year is slightly decreased as compared to 1975. In other words, the comparison between 1975 and 2000 can better explain the conclusion of this paper. The global average rNc in 2000 (1.075) is also slightly less than that in 1975 (1.076). For the period from 1975 to 2000, the spatial pattern of anthropogenic AOD and rNc is noticeably different. Anthropogenic aerosol pollution decreases over Europe and North America but increases over East and South Asia.

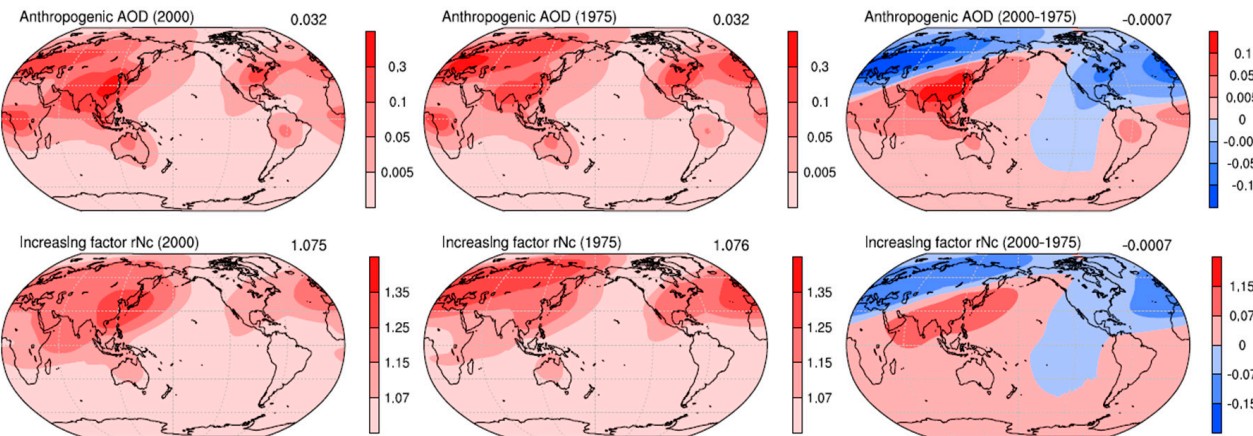

**Figure 7.** Anthropogenic aerosol optical depth in the visible band (AOD, upper panels) and the increasing factor for cloud droplet number (rNc, lower panels) provided by MACv2-SP for 2000 (**left**), 1975 (**middle**), and their difference (**right**). The global average is given in the upper-right corner.

$\Delta F_{AaCc} = \Delta(F_{AaCc} - F_{ACc}) + \Delta(F_{ACc} - F_{AC}) + \Delta F_{AC} = \Delta aF_{ACc} + \Delta cF_{AC} + \Delta F_{AC}$. In other words, the anthropogenic aerosol ERF ($\Delta F_{AaCc}$) can be decomposed into the RFari ($aF_{ACc}$), RFaci ($cF_{AC}$), and rapid adjustment ($\Delta F_{AC}$). Figure 8 shows comparisons of RFari ($aF_{ACc}$ and $aF_{Cc}$), RFaci ($cF_{AC}$), and rapid adjustment ($\Delta F_{AC}$) between the ALL (2000) and PAT (1975) experiments. From 1975 to 2000, both RFari and RFaci decrease (less negative) in Europe and North America but increase (more negative) in East and South Asia. This result is in agreement with the spatial changes in anthropogenic AOD and rNc (Figure 7). In terms of the global average, both $aF_{ACc}$ and $aF_{Cc}$ grew stronger (more negative) by $-0.015$ W m$^{-2}$ from 1975 to 2000. These changes are very robust because the estimated $aF_{ACc}$ and $aF_{Cc}$ are scarcely affected by model internal year-to-year variability. Notably, this result is inconsistent with the fact that the global average AOD slightly decreased from 1975 to 2000 (Figure 7). Furthermore, the global average $cF_{AC}$ increases (more negative) by $-0.008$ W m$^{-2}$ from 1975 to 2000, which is also inconsistent with the trend of the global average rNc that slightly decreased from 1975 to 2000 (Figure 7). These results suggest that under the same global mean anthropogenic AOD and rNc values, changing anthropogenic aerosol spatial patterns have a clear impact on the global average RFari and RFaci. From 1975 to 2000, with major anthropogenic aerosol emissions occurring in East and South Asia instead of in Europe and North America, the global average anthropogenic aerosol RF (RFari + RFaci) enhanced (more negative) by ~6%. Each of $F_{AC}^{ALL–BASE}$, $F_{AC}^{PAT–BASE}$ and $F_{AC}^{ALL–PAT}$ show that rapid adjustments are obviously affected by model internal year-to-year variability. Therefore, it is difficult to detect the change in the global average anthropogenic aerosol ERF (i.e., RF + rapid adjustments) caused by the spatial shift in anthropogenic aerosol. This is the reason why the EOF was decomposed into robust RF and unstable rapid adjustments in this section.

In order to determine why the global average anthropogenic aerosol RF is enhanced (more negative) even with a slight decrease in anthropogenic aerosol, the corresponding efficiencies are shown in Figure 9. The efficiencies from the ALL (2000) experiment are generally similar to those from the PAT (1975) experiment. Compared to the efficiencies in 2000, some high-efficiency areas over the Pacific Ocean in 1975 were masked, owing to a low anthropogenic aerosol burden (AOD < 0.005 or rNc < 1.07). Regionally, the radiative efficiencies for $aF_{ACc}$, $aF_{Cc}$, and $cF_{AC}$ over East and South Asia and their adjacent oceans are generally more negative than those over Europe and North America. This is the reason why, for the period from 1975 to 2000, the global average RF (RFari + RFaci) is enhanced (more negative) by ~6%, even with a slight decrease in anthropogenic aerosols.

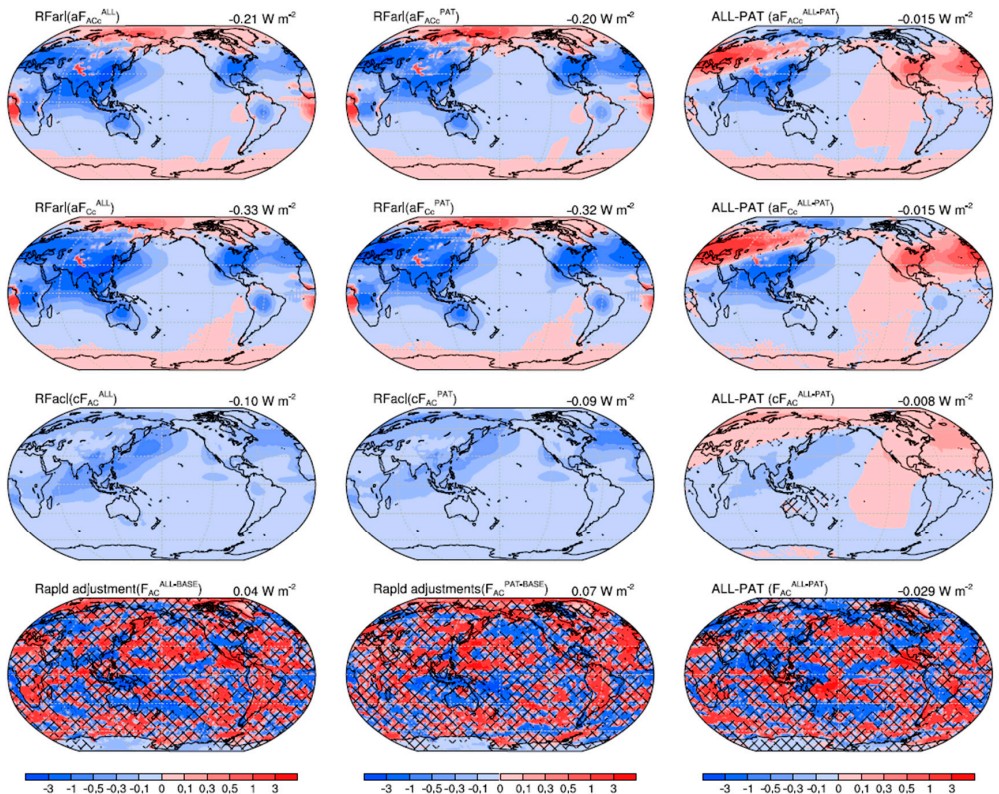

**Figure 8.** Annual mean maps for the three basic components of anthropogenic aerosol effective radiative forcings, i.e., the instantaneous direct radiative effect (RFari, aFACc and aFCc; top two rows), instantaneous Twomey effect (RFaci; third row), and rapid adjustment (bottom row) with anthropogenic aerosol data for 2000 (**left**), 1975 (**middle**), and their difference (**right**). The global average is given in the upper-right corner. Hatching represents the nonsignificant area at the 90% confidence level of Student's *t*-test.

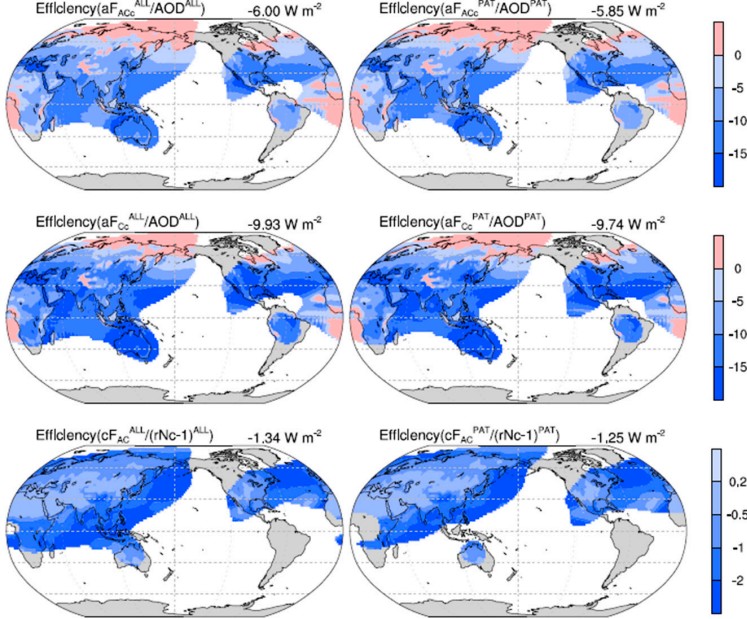

**Figure 9.** Annual means of the two radiative efficiencies for the anthropogenic aerosol direct radiative effect with and without the influence of natural aerosol (aF$_{ACc}$/AOD and aF$_{Cc}$/AOD, **top two** rows) and the efficiency for the Twomey effect (cF$_{AC}$/(rNc − 1), **bottom** row) from the ALL (2000, **left**) and PAT (1975, **right**) experiments. The area with a low anthropogenic aerosol burden (AOD < 0.005 or rNc < 1.07) is masked.

## 4. Conclusions and Discussion

In this study, the prescribed anthropogenic aerosol forcing recommended by CMIP6 was implemented in the GAMIL model. Although the anthropogenic aerosol radiative effects were estimated, we have not paid attention to the similarities and differences between our estimates and those reported in other studies. This study focuses on how to take full advantage of the prescribed anthropogenic aerosol forcing.

With reduced complexity of anthropogenic aerosol forcing, each component of the anthropogenic aerosol ERF can be estimated, and the RFari and RFaci can be estimated by all possible calculation methods. Simulation results show that both RFaci and RFari from a present-day simulation (i.e., double radiation calls) are very robust. These RFaci (e.g., $cF_{AC}^{ALL}$) and RFari (e.g., $aF_{ACc}^{ALL}$) are scarcely affected by model internal year-to-year variability and can be used as offline simulation results. However, the anthropogenic aerosol RFari determined by the difference in total aerosol RFari between preindustrial and present-day simulations (e.g., $AaF_{Cc}^{ALL-BASE}$, a commonly used method) is obviously affected by model internal year-to-year variability. This suggests that, if possible (e.g., using prescribed anthropogenic aerosol forcing), it is preferable to diagnose the RFari at each model time step. Simulation results also show that the impact of natural aerosols on calculating RFari is notable. Because the preindustrial aerosol (i.e., natural aerosol) level used for climate models is full of uncertainty [28], the impact of natural aerosols on calculating RFari might differ widely among climate models. If possible, it is also necessary to compare the RFari without the influence of natural aerosols. The RFaci—calculated by double radiation calls—is the definition-based Twomey effect (i.e., instantaneous Twomey effect), which was previously impossible to diagnose using the default model with physically based aerosol–cloud interactions. More importantly, the perturbation of the ERF depends almost entirely on its rapid adjustment (i.e., ERF − RF). If possible, it is better to decompose the ERF into stable components (i.e., RFari and RFaci) and unstable components (i.e., rapid adjustments).

In terms of the robust RFari and RFaci, the anthropogenic aerosol radiative effects can be estimated from various other perspectives. For instance, the seasonal variability of RFari and RFaci can be analyzed. The seasonal variability of the global average RFaci is much stronger than that of RFari. This can be explained by the corresponding radiative forcing efficiency, which indicates the impact of environmental properties (such as surface albedo, solar insolation and even clouds). Another example is the impact of the spatial shift in anthropogenic aerosol. For the period from 1975 to 2000, anthropogenic aerosol pollution decreased over Europe and North America but increased over East and South Asia. Excluding the unstable components of the ERF (i.e., rapid adjustments), its stable components (i.e., RFari and RFaci) show a clear response to the spatial shift in anthropogenic aerosols. The radiative forcing efficiencies for RFari and RFaci over East and South Asia and its adjacent oceans are generally stronger than those over Europe and North America, especially for RFaci. As a result, from 1975 to 2000, the global average RF (RFari + RFaci) was enhanced (more negative) by ~6%, even with a slight decrease in the global average anthropogenic aerosols. In short, the robust RFari, RFaci, and corresponding efficiencies are very useful for analyzing anthropogenic aerosol radiative effects.

One should keep in mind that the conclusion, which comes from prescribed anthropogenic aerosol forcing, omits the coupling between synoptic systems and anthropogenic aerosol. For instance, rainier days are cloudier, but also have lower aerosol levels. If considering the rainy effect, over Southeast Asia, the RFaci and its efficiency might not be so strong in summer, which is also the rainy season (Figure 6). On the other hand, it is difficult for climate models with physically based anthropogenic aerosol processes to directly calculate the robust components of the ERF (i.e., RFari and RFaci). Furthermore, it is clear that the efficiency is dependent on the modeled cloud properties. Because the differences in cloud properties among climate models are notable, the spatial pattern and seasonal variability of efficiency from the GAMIL model might be different from other models. The

differences in estimating CMIP6 anthropogenic aerosol radiative effects among climate models can be explained by the differences in efficiency.

**Author Contributions:** X.S. designed the study and wrote the original draft. X.S. and L.L. developed the model code; C.L. contributed to some writing of the manuscript. W.Z. and J.L. contributed to review and editing of the manuscript. All authors have read and agreed to the published version of the manuscript.

**Funding:** This research was funded by the National Key Research and Development Program of China (grant nos. 2018YFC1507001) and the National Natural Science Foundation of China (grant nos. 41775095). The APC was funded by the same funding.

**Data Availability Statement:** Model source code and simulation results used in this study have be archived in a public repository http://doi.org/10.5281/zenodo.3516714 (Accessed on 7 January 2021).

**Acknowledgments:** The authors would like to thank Bjorn Stevens for guidance in using MACv2-SP. The analysis of model results was conducted in the High-Performance Computing Center of Nanjing University of Information Science & Technology.

**Conflicts of Interest:** The authors declare no conflict of interest.

**Appendix A**

Table A1 lists the global annual mean radiative variables from all experiments. As a supplement to the main text, this section introduces all possible calculation methods for estimating anthropogenic aerosol radiative effects based on the simulation results listed in Table A1. Meanwhile, the differences among these calculation methods are discussed.

**Table A1.** Global annual means of the variables listed in Tables 1 and 2. The standard deviations (in brackets) are calculated from the difference of each year for 10 years. All values are in the unit of W m$^{-2}$.

| | BASE | RAD−BASE | ALL–TMY | TMY−BASE | ALL−RAD | ALL−BASE | PAT−BASE |
|---|---|---|---|---|---|---|---|
| $F_{AaCc}$ | 237.96 | −0.22 (0.17) | −0.09 (0.15) | −0.19 (0.17) | −0.05 (0.16) | −0.27 (0.13) | −0.22 (0.15) |
| $F_{AC}$ | 237.96 | −0.01 (0.17) | 0.12 (0.15) | −0.09 (0.17) | 0.05 (0.16) | 0.04 (0.13) | 0.07 (0.15) |
| $F_A$ | 285.18 | 0 (0.05) | 0.01 (0.07) | −0.02 (0.07) | −0.01 (0.05) | −0.01 (0.05) | 0.03 (0.05) |
| $F$ | 293.75 | 0 (0.06) | 0.01 (0.07) | −0.02 (0.07) | −0.01 (0.06) | −0.01 (0.05) | 0.03 (0.05) |
| $AaF_{Cc}$ | −5.74 | −0.21 (0.01) | −0.22 (0.01) | 0.01 (0.01) | 0 (0.01) | −0.21 (0.01) | −0.20 (0.01) |
| $AaF$ | −8.58 | −0.45 (0.01) | −0.45 (0.01) | 0 (0) | 0 (0) | −0.45 (0.01) | −0.43 (0.01) |
| $aF_{ACc}$ | | −0.21 (0) | −0.21(0) | | 0 (0) | −0.21 (0) | −0.20 (0) |
| $aF_A$ | | −0.45 (0) | −0.45 (0) | | 0 (0) | −0.45 (0) | −0.42 (0) |
| $aF_{Cc}$ | | −0.33 (0) | −0.33 (0) | | 0 (0) | −0.33 (0.01) | −0.32 (0) |
| $aF$ | | −0.64 (0) | −0.64 (0) | | 0 (0) | −0.64 (0) | −0.61 (0) |
| $aF_{AC}$ | | −0.21 (0) | −0.21 (0) | | 0 (0) | −0.21(0) | −0.20 (0) |
| $aF_{Cc}dA$ | | 0.12 (0) | 0.12 (0) | | 0 (0) | 0.12 (0) | 0.12 (0) |
| $aFdA$ | | 0.19 (0) | 0.19 (0) | | 0 (0) | 0.19 (0) | 0.19 (0) |
| $CcF_{Aa}$ | −47.22 | 0.23 (0.18) | 0.36 (0.15) | −0.17 (0.16) | −0.04 (0.17) | 0.19 (0.13) | 0.17 (0.15) |
| $CcF$ | −50.05 | −0.01 (0.19) | 0.12 (0.15) | −0.17 (0.17) | −0.04 (0.18) | −0.05 (0.14) | −0.06 (0.15) |
| $CcF_A$ | −47.22 | −0.01 (0.18) | 0.12 (0.15) | −0.17 (0.16) | −0.04 (0.17) | −0.05 (0.13) | −0.05 (0.14) |
| $CF_{Aa}$ | −47.22 | 0.23 (0.18) | 0.35 (0.15) | −0.07 (0.16) | 0.06 (0.17) | 0.29 (0.13) | 0.27 (0.14) |
| $CF_A$ | −47.22 | −0.01 (0.18) | 0.12 (0.15) | −0.07 (0.16) | 0.06 (0.17) | 0.05 (0.13) | 0.04 (0.14) |
| $cF_{AaC}$ | | | 0 (0) | −0.10 (0) | −0.10 (0) | −0.10 (0) | −0.09 (0) |
| $cF_{AC}$ | | | 0 (0) | −0.10 (0) | −0.10 (0) | −0.10 (0) | −0.09 (0) |

There is an excess of methods for calculating RFari, and these methods can be classified into two categories. Firstly, the RFari can be diagnosed as the difference in all-sky total aerosol forcing between two simulations with and without anthropogenic aerosols ($\Delta AaF_{Cc}$, $AaF_{Cc} = F_{AaCc} - F_{Cc}$). This method is the commonly used method introduced by Ghan (2013), and can be employed in climate models with physically based anthropogenic aerosol processes (i.e., anthropogenic aerosol mixed with natural aerosol) [9]. Using this method, the RFari can be quantified by $AaF_{Cc}^{RAD-BASE}$, $AaF_{Cc}^{ALL-TMY}$, and

$AaF_{Cc}^{ALL-BASE}$. The differences between these three variables are small and not significant. The clear-sky RFari (i.e., $\Delta AaF$) can be quantified by $AaF^{RAD-BASE}$, $AaF^{ALL-TMY}$, and $AaF^{ALL-BASE}$. The differences between these three variables are also small and not significant. Secondly, the RFari can be obtained from a simulation with prescribed anthropogenic aerosol optical properties (e.g., the ALL or RAD experiment). Based on the ALL or RAD experimental results, both $aF_{ACc}$ ($aF_{ACc} = F_{AaCc} - F_{ACc}$), $aF_{Cc}$ ($aF_{Cc} = F_{aCc} - F_{Cc}$), and $aF_{AC}$ ($aF_{AC} = F_{AaC} - F_{AC}$) can represent the all-sky RFari. The $aF_{Cc}$ is obviously more negative than the $aF_{ACc}$ because the impact of natural aerosols on calculating the all-sky RFari is removed. The difference between $aF_{ACc}$ and $aF_{AC}$ shows the impact of the Twomey effect on calculating RFari, which is negligible. Both $aF_A$ ($aF_A = F_{Aa} - F_A$) and $aF$ ($aF = F_a - F$) can represent the clear-sky RFari. The obvious difference between $aF_A$ and $aF$ (i.e., $aFdA = aF_A - aF$) indicates the impact of natural aerosols on calculating the clear-sky RFari. It is noteworthy that all these anthrophonic aerosol direct radiative forcing variables (i.e., aFxx, where the subscript "XX" refers to background radiative forcing factor) from the RAD experiment are almost the same as those from the ALL experiment. This also suggests that the impact of the anthropogenic aerosol Twomey effect (i.e., the difference between the RAD and ALL experiments) on estimating RFari is negligible.

The anthropogenic aerosol indirect effects on warm clouds are often estimated by their impact on shortwave cloud forcing, which is the difference in $CcF_{Aa}$ ($CcF_{Aa} = F_{AaCc} - F_{Aa}$), $CcF_A$ ($CcF_A = F_{ACc} - F_A$) or $CcF$ ($CcF = F_{Cc} - F$) between two simulations with and without anthrophonic aerosol. Ghan (2013) pointed out that $\Delta CcF_{Aa}$ is positively biased due to the impact of the anthrophonic aerosol direct radiative effect [9]. This is the reason why $CcF_{Aa}^{ALL-BASE}$ is obviously larger than $CcF_A^{ALL-BASE}$ and $CcF^{ALL-BASE}$. Because the TMY experiment does not consider the anthrophonic aerosol direct radiative effect, $CcF_{Aa}^{TMY-BASE}$ is the same as $CcF_A^{TMY-BASE}$, and they are close to $CcF^{TMY-BASE}$. In terms of definition, the Twomey effect is the instantaneous radiative forcing (i.e., RFaci). The aerosol indirect effects estimated by $\Delta CcF$ and $\Delta CcF_A$ are not the exact Twomey effect because of the subsequent changes in cloud forcing from rapid adjustments. Both $CcF_A^{TMY-BASE}$ and $CcF_A^{ALL-RAD}$ include the rapid adjustments in cloud forcing induced by the Twomey effect. Compared to $CcF_A^{TMY-BASE}$, $CcF_A^{ALL-BASE}$ also includes rapid adjustments in cloud forcing induced by the anthrophonic aerosol direct radiative effect (i.e., semi-direct effect). With MACv2-SP, the RFaci can be calculated by double radiation calls at each radiation time step. Based on the ALL or TMY experimental results, both $cF_{AaC}$ ($cF_{AaC} = CcF_{Aa} - CF_{Aa} = F_{AaCc} - F_{AaC}$) and $cF_{AC}$ ($cF_{AC} = CcF_A - CF_A = F_{ACc} - F_{AC}$) can represent RFaci (i.e., the definition-based Twomey effect). The comparison between $cF_{AaC}$ and $cF_{AC}$ indicates that the impact of the anthropogenic aerosol direct radiative effect on estimating RFaci is negligible. The difference between the anthropogenic aerosol indirect effects (i.e., the Twomey effect and corresponding rapid adjustments) estimated by $\Delta CcF_A$ (e.g., $CcF_A^{TMY-BASE}$) and the Twomey effect estimated by $cF_{AC}$ (e.g., $cF_{AC}^{TMY}$) is the rapid adjustments in cloud forcing induced by the Twomey effect, which cannot be ignored. In other words, even if the lifetime effect is excluded, the definition-based Twomey effect cannot be approximated by the difference in cloud forcing between two simulations with and without the Twomey effect. Finally, it is necessary to point out that the modeled RFaci (i.e., $cF_{AaC}$ or $cF_{AC}$) from the TMY experiment is the same as that from the ALL experiment. This also suggests that the impact of the anthropogenic aerosol direct radiative effect (i.e., the difference between the TMY and ALL experiments) on estimating RFaci is negligible. The impact of the anthropogenic aerosol direct radiative effect on estimating the aerosol indirect effects on warm clouds (e.g., $CcF_{Aa}^{ALL-BASE} - CcF_A^{ALL-BASE}$) is obvious, owing to the semi-direct effect. Attention should be paid to this difference.

This paragraph introduces the calculation methods for estimating the anthropogenic aerosol ERF, RF and rapid adjustments (ERF − RF). Unlike RF, the ERF and rapid adjustments must be estimated by the difference between two simulations with and without anthropogenic aerosol forcing. The anthropogenic aerosol ERF is calculated as $\Delta F_{AaCc}$. With the benefit of MACv2-SP, $\Delta F_{AaCc}$ can be decomposed into $\Delta(F_{AaCc} - F_{AC})$ and $\Delta F_{AC}$.

It should be noted that the $F_{AaCc} - F_{AC}$ from the simulation without anthropogenic aerosol is zero and the $F_{AaCc} - F_{AC}$ from the simulation with anthropogenic aerosol represents RF (i.e., RFari + aci). Thus, $\Delta F_{AC}$ indicates the rapid adjustments. The ERFari + aci is quantified by $F_{AaCc}^{ALL-BASE}$. The corresponding rapid adjustment is quantified by $F_{AC}^{ALL-BASE}$. Both RFari ($aF_{ACc} = F_{AaCc} - F_{ACc}$, $aF_{Cc} = F_{aCc} - F_{Cc}$, or $aF_{AC} = F_{AaC} - F_{AC}$) and RFaci ($cF_{AaC} = F_{AaCc} - F_{AaC}$ or $cF_{AC} = F_{ACc} - F_{AC}$) can be calculated based on the ALL experiment. It should be noted that RFari + RFaci may not be equal to RFari + aci ($F_{AaCc}^{ALL} - F_{AC}^{ALL}$), except for $aF_{AC}^{ALL}$ (RFari) + $cF_{AaC}^{ALL}$ (RFaci) = $aF_{ACc}^{ALL}$ (RFari) + $cF_{AC}^{ALL}$ (RFaci) = $(F_{AaCc} - F_{AC})^{ALL}$ (RFari + aci). Because the rapid adjustments from aerosol–radiation interactions and the rapid adjustments from aerosol–cloud interactions will be mixed at each time step in the simulation, it is impossible to separate them in the ALL experiment. Another simulation, which switches off the Twomey effect or direct radiative effect is needed for estimating the rapid adjustments from the direct radiative effect or Twomey effect. Both $F_{AC}^{RAD-BASE}$ and $F_{AC}^{ALL-TMY}$ can represent the rapid adjustment from aerosol–radiation interactions. The $\Delta F_{AC}$ can be decomposed into $\Delta CF_A$ ($CF_A = F_{AC} - F_A$) and $\Delta F_A$. The $\Delta CF_A$ ($CF_A^{RAD-BASE}$ or $CF_A^{ALL-TMY}$) indicates the anthropogenic aerosol semi-direct effect. The absolute value of the global average $\Delta F_A$ ($F_A^{RAD-BASE}$ or $F_A^{ALL-TMY}$) is usually very small, and therefore, neglectable. As such, the anthropogenic aerosol rapid adjustments from aerosol–radiation interactions are often approximated by the semi-direct effect [8]. Both $F_{AC}^{TMY-BASE}$ and $F_{AC}^{ALL-RAD}$ can represent the rapid adjustment from aerosol–cloud interactions. It is noteworthy that the year-to-year variability (standard deviation in brackets) of rapid adjustments (i.e., $\Delta F_{AC}$) is obviously larger than the annual mean. This is the reason why more attention has been paid to the stable RFari and RFaci in this study. The 10-year simulation is sufficient for calculating the stable RFari and RFaci.

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
