# Peer review of "Estimating the CMIP6 Anthropogenic Aerosol Radiative Effects with the Advantage of Prescribed Aerosol Forcing"

_atmosphere, doi:10.3390/atmos12030406_

Round 1
Reviewer 1 Report
Review of “Estimating the CMIP6 anthropogenic aerosol radiative effects with the advantage of prescribed aerosol forcing” by Shi et al.
The paper describes estimated anthropogenic aerosol radiative effects using prescribed aerosol forcing. Overall, the paper is well written and mostly clear, and I found the approach and results to be interesting and instructive. I would suggest publication after the comments below are addressed.
General comments
- The work considers only shortwave radiation and, I believe only fine particles. Can the authors discuss these caveats some more, especially in the conclusion section? What might be the effect of this limitation? And why is this assumption made?
- It is clear here what the advantage of using prescribed aerosol properties is. However, one could argue that is a highly idealized model and it is not clear how that assumption might affect the results. Can the authors discuss this caveat some more? Could one still use prescribed aerosol properties to determine all the differences and separate out all the different effects, but also compare the overall forcing with a fully physically based (no prescribed properties) model to see how realistic the numbers are in that scenario at least? That would give more credibility to the number derived from the prescribed model for the differences.
- The use of so many acronyms for the different runs and differences makes the read a bit difficult at times, but I don’t have a suggestion on how to improve that, and I would say, the authors did a good job in trying to make the nomenclature, for the most part, as intuitive as might be possible.
Specific comments
Line 10: spell out the CMIP6 acronym.
Line 15: “…by double radiation calls…” I am not clear what that means.
Line 22: “…is enhanced…” meaning increases in the positive direction, or meaning more negative, or something else?
Line 33: “idealized experiments” refers to numerical simulation experiments, thought experiments, or actual experiments?
Line 37: “prescribed” in what sense?
Line 44: Explain what “double radiation calls” indicates. That becomes somewhat clear later, but here and in the abstract leaves the reader guessing.
Lines 63 to 65: These two sentences, as written seem to contradict each other; if RF contributes most of the ERF and RF is stable, why would ERF vary considerably?
Line 74: add “a” in front of “previous”.
Line 75: “diverse” in what sense?
Line 83: Add “the” in front of “Chinese”
Line 105: The host model mechanism” is a bit vague.
Lines 105-107: Why only fine aerosols are considered to contribute, and what is the effect of excluding coarse mode particles? It would be good to clearly define that ratio in an equation.
Line 108: why the “r” in the acronym “rNc”? Is that for “ratio”? If so, maybe instead of “change” here use “ratio”.
Line 110: remove “works”. It is confusing in my opinion, either leave it without the verb or change it for “contributions” or “contributes”.
Line 114: What is the justification to include only the effect on warm clouds?
Lines 137-138: As already mentioned before, please explain what “double radiation calls” means and how it is implemented. I think this is explained in the paragraph starting at line 155, but this should be made clear (although succinctly) much earlier on starting with a brief explanation already in the abstract, and then repeated early in the introduction, etc. Otherwise, the reader is left guessing until halfway through the paper.
Line 153: “corresponding rapid adjustments”, such as? For the rapid adjustments contributing to ERFari, the authors provided the example of the semidirect effect, can the authors also provide an example also for the rapid adjustments’ contribution to ERFaci?
Line 162: “indicates” should be “indicate”
Table 1 and related text: I am confused by FCc, if I understand this correctly, this should be the forcing including the Twomey effect without the natural and without the anthropogenic aerosols (meaning no aerosol at all?). How can there be a Twomey effect without aerosols (neither anthropogenic nor natural)? I think this indicates the forcing associated with the Twomey effect of (all?) aerosols but excluding the aerosol direct radiative effect. This seems also to be confirmed by Table 2. Is that correct? It might be good to clarify this point in the paragraph starting on line 155 or the following starting on line 174.
The paragraph starting at line 190: I am curious what does the acronym PAT stay for (pre-something-something)?
Line 205; Is it the standard deviation or the average? Or is it the standard deviation of the average (i.e., the standard error)?
Line 239: “pointed” should be “point”
Line 244: I see how the absorption would matter here (above bring surfaces), a bit less the scattering. Can the authors elaborate some more?
Lines 255 and 256: What does the “d” stay for in aFCcdA abd aFdA? I guess this is explained in line 259, but I think this should be explained immediately before the sentence currently in lines 255-256.
Line 265: Even if this impact might be “obvious”, I still think it is quite important.
Line 288: “pointed” -> “point”
Lines 309 – 310: Can the authors provide some explanations/examples of how the instantaneous Twomey effect differs from the subsequent forcing induced by it?
Line 343: I understand the intentions of the authors, by “weakest” can be confusing because these numbers are negative, so a more negative number might seem like should be weaker than a less negative one, which is the opposite of what is meant here, maybe just say “less negative”?
Line 368: consider removing “the” before “land”
Line 370: Also here, “enhances” can be confusing, enhances in an absolute value sense (negative to more negative, and positive to more positive) or not (e.g., negative to less negative)? Maybe better to be explicit.
Lines 387-388: I am not sure I fully grasp the explanation for why choosing 2000.
Line 401: Similarly, the use of the term “decrease” as “more positive” is confusing, I would just use “more positive”
Line 404: Ditto (increases -> more negative)
Line 408: Ditto.
Line 427: Ditto (enhanced)
Line 433: Ditto. Stronger in which sense (more negative? Or less negative?)?
Line 446: Why not paying attention?
Line 483: Again “enhanced”, clarify.
Line 488: Maybe add “with” in front of “less”
Line 532: “stronger” -> more negative…
Table A1: Are these normalized values or are they in W/m2? If not normalized, then please provide units in the table.
Reviewer 2 Report
The manuscript uses prescribed anthropogenic aerosol forcing from the MACv2-SP simple plume model, which is used in Coupled Model Intercomparison Project Phase 6 29 (CMIP6) climate simulations, within the Grid-point Atmospheric 81 Model of IAP LASG (GAMIL) model to simulate present day (1975 and 2000) direct and indirect responses to anthropogenic aerosol forcing. The manuscript uses differences between fixed SST experiments with and without aerosol direct and indirect effects to determine effective radiative forcing (ERF) and uses multiple calls to the model radiation schemes with and without anthropogenic direct and indirect aerosol effects to determine instantaneous radiative forcing (RF). The rapid adjustment response is estimated as a difference ERF-RF.
The manuscript discusses the sensitivity of the ERF estimates to different approaches for calculating it, describes the seasonal variability in RF, and also looks at the impact of shifts in anthropogenic aerosols on the radiative forcing. The manuscript does a very good job of defining the different short wave fluxes (Table 1) and RF diagnostics (Table 2) used in the experiments. The experiments are also clearly outlined. This is critical, since the nomenclature for describing the experiments is extremely complicated.
For this reviewer, the results of the rapid response residual diagnostics show no significance and very little spatial structure, seems that these residual calculations are not particularly robust. This should be emphasized in the conclusions. Otherwise, the research reported in the manuscript is highly relevant and I recommend publication of the manuscript after minor concerns are addressed.
Line 99: Add discussion of fixed aerosol plumes from specific point sources and imposed vertical and seasonal cycles used in the MACv2-SP model.
Line 203: Please clarify, 2.5x2 degrees in lat/lon?
Line 204: Please clarify, model top, number of tropospheric and stratospheric layers?
Line 231-232: This is not a difference, so how could it have statistical significance?
Line 239: “pointed” should be “point”
Lines 305-306: What is the motivation for this decomposition? It’s not clear in the discussion.
Lines 325-326: Not clear what the conclusion of this discussion is. Neither the Twomey effect (upper panels of Figure 4) or the rapid adjustment (lower panels of Figure 4) show any significant spatial structure and are largely insignificant.
Line 400: Most of the rapid adjustment diagnostics in Figure 8 show low significance and very little spatial structure, seems that these residual calculations are not particularly robust. This should be emphasized in the conclusions.
Reviewer 3 Report
Presented paper looks important and works with prescribed anthropogenic aerosol forcing recommended by CMIP6. This task has direct connection with the problem of climate change. This aerosol implemended in an atmospheric model. The results of model simulations showed that the global averaged RF increase from 1975 to 2000 up to 6%. Possibly this result is important for our understanding radiation-aerosol problem, but it may be corrected if we will use another model with improved cloud-aerosol-radiation moduls. These sheems usually are very complicated as we know.
So more details about the models which has been used
( upper boundary ets ) would be useful.
